# The Relation between Drying Conditions and the Development of Volatile Compounds in Saffron (*Crocus sativus*)

**DOI:** 10.3390/molecules26226954

**Published:** 2021-11-18

**Authors:** Teresa Soledad Cid-Pérez, Guadalupe Virginia Nevárez-Moorillón, Carlos Enrique Ochoa-Velasco, Addí Rhode Navarro-Cruz, Paola Hernández-Carranza, Raúl Avila-Sosa

**Affiliations:** 1Departamento de Bioquímica-Alimentos, Facultad de Ciencias Químicas, Benemértia Universidad Autónoma de Puebla, Puebla 72240, Mexico; teresa.cid@correo.buap.mx (T.S.C.-P.); carlos.ochoa@correo.buap.mx (C.E.O.-V.); addi.navarro@correo.buap.mx (A.R.N.-C.); paola.hernandezc@correo.buap.mx (P.H.-C.); 2Facultad de Ciencias Químicas, Universidad Autónoma de Chihuahua, Circuito Universitario s/n Campus Universitario II, Chihuahua 31125, Mexico; vnevare@uach.mx

**Keywords:** saffron, flavor and odor pathways, volatile compounds

## Abstract

Saffron is derived from the stigmas of the flower *Crocus sativus* L. The drying process is the most important post-harvest step for converting *C. sativus* stigmas into saffron. The aim of this review is to evaluate saffron’s post-harvest conditions in the development of volatile compounds and its aroma descriptors. It describes saffron’s compound generation by enzymatic pathways and degradation reactions. Saffron quality is described by their metabolite’s solubility and the determination of picrocrocin, crocins, and safranal. The drying process induce various modifications in terms of color, flavor and aroma, which take place in the spice. It affects the aromatic species chemical profile. In the food industry, saffron is employed for its sensory attributes, such as coloring, related mainly to crocins (mono-glycosyl esters or di-glycosyl polyene).

## 1. Introduction

Plant extracts can be sources of valuable chemical compounds, with particular biological properties, such as color or aroma. Tinctures or essential oils derived from plants are usually a complex mixture of bioactive components, where their combination usually present synergistic effects. Due to their aromatic nature and the biological activity of their components, spices have been valued for centuries, and among them, saffron is one of the most precious.

Saffron spice is made up of the dried stigmas of the *Crocus sativus* L. flower. Its primary use is in food, where it is valued for its coloring, flavoring, and aromatizing of some traditional dishes [1]. Yield loss can be caused by several factors, including poor soil fertility, the lack of assured irrigation and available high-quality corms as propagation material, and infestation by rodents and diseases including *Fusarium oxysporum, Penicillium* spp., and *Rhizoctonia violacea*. Other factors are inadequate postharvest management, improper marketing facilities, adulteration, and the adverse effect of climate change [2]. The post-harvest drying process represents a critical step in the development of saffron, with specific physical and chemical morphological characteristics [3,4]. Reductions in the commercial quality of saffron can be attributed to inappropriate harvesting methods, insufficient dehydration processing, exposure to direct sunlight, improper storage, and adulteration [5,6].

Saffron is employed in the confectionery, gastronomic, and liquor industries [4,7,8]; its quality is described by the solubility and concentration of its metabolites. *C sativus* is the only plant species that produces substances such as crocins, safranal and picrocrocins, which are derived from primary metabolites in significant amounts [9,10]. Safranal is the key component of saffron essential oil and is responsible for its odor. Crocin contributes to color and is widely used in the food industry, pharmaceutical, cosmetics and perfumery industries, as well as in the production of textile dyes [10,11]; meanwhile, picrocrocin is considered to be the principal contributor of its bitter taste [12,13]. The aim of this review is to evaluate saffron’s post-harvest conditions in the development of volatile compounds and its aroma descriptors.

## 2. Saffron as an Aromatic Spice

Saffron is derived from the stigmas of the flower *Crocus sativus* L. The stigmas are dried and used as a natural dye or as flavoring in cooking [14,15]. In addition to providing color to food, it acts as an antioxidant [16] and has pharmacological properties [17]. The name saffron is derived from the Arabic za’-faran (“yellow”) or from the Persian term Za’afarn, which means golden flowers. The Greeks knew it as *krokos* [8,18], since mythology described how the god Hermes (Mercury) wounded his friend Krokos in the head; when he fell dead, his blood spilled onto a flower, creating three blood-colored filaments [19]. Historically, the cultivation and use of saffron has spanned more than 3500 years across multiple cultures, continents, and civilizations [20]. The origin of the practice is believed to be in the eastern Mediterranean basin, from which farming of the plant spread to other parts the “Old World”. Many species of the *Crocus* are obtained from Crete and islands in the Aegean sea, which can be considered as the "birthplace" of saffron [17]. Saffron is among the most traditional spices, and even appeared in a fresco from the Palace of Minos in Crete. Today, the spice it is known as “red gold” or “soft gold” [21,22,23]. 

### 2.1. Botanical Characterisctics

*C. sativus* L. is a triploid plant, geophyte of the Iridaceae family that blooms in autumn and grows in arid and semi-arid climates [24,25]. It consists of a corm, dark green leaves, and lilac flowers (Figure 1). Corms produce symmetrical white roots covered with several tunics of varying texture and color [26,27,28]. The plant can bloom with one or several flowers whose peduncle and ovary are underground. The flower has a perigonium consisting of six purple tepals (2–4.7 cm long and 1.1 and 2.3 cm wide), three stamens, and stigma. The stigma is arranged by three red filaments linked by their base to the style, with an enlarged apex in the form of a trumpet. In the anthesis, these elements remain upright, but as the flower opens, the tepals tilt downward from longer filaments than the anthers [2,7,27,29]. Sub-histerant flowers (appearing before or after the leaves) are sterile, so the plant propagates through underground corms from which flowers emerge in autumn [2,26,28,30]. The life cycle of *C. sativus* is composed of two stages: vegetative (to propagate corms) and reproductive (formation of floral organs) [30]. The flower number, size, and stigma are conditioned by corm characteristics [29]; however, stigma length and color are related to the development and plant growth. Some authors [4,31] have described the stigma color as changing from scarlet-yellow in the following phases: yellow stigma, closed bud inside the perianth tube (0.3 cm in length); orange stigma, closed bud inside the perianth tube (0.4 cm in length); red stigma, closed bud inside the perianth tube (0.8 cm in length); three days before anthesis (complete flower opening), dark red stigmas in closed bud outside the perianth tube (3 cm); day of anthesis, dark red stigmas form in the closed bud outside of the perianth tube (3 cm); on the day of anthesis, the dark red stigmas are 3 cm in length, and they last until two days after anthesis. Stigma coloration changes are associated with the increase of carotenoids and biosynthesis, and the increase and accumulation of apocarotenoids and zeaxanthin derivatives (crocin, crocetin and picrocrocin) [32].

The plant is distributed geographically around the world, but production is concentrated in Iran. However, significant producers include India, Greece, Afghanistan, Morocco, and Spain. The world production of saffron is approximately 430 tons per year [25,33], with the principal importing countries being Spain (23%), Hong Kong (8.7%), the United States (7.6%), India (7.0%), and Italy (7.0%) [2].

### 2.2. Harvest

*C. sativus* farmers have observed certain advantages of planting, such as high market prices, low water requirements, and a long-term exploitation opportunity with only a single crop [34]. To produce 1 kg of saffron, approximately 150,000 flowers must be collected within 370–470 h (i.e., the flowers must be picked, their stigmas removed, and the drying process conducted) [26,28]. *C. sativus* occupies a very special place, since it is the only plant species that naturally produces crocins, safranal, and picrocrocins in significant quantities [10]. Safranal is the main component of saffron essential oil and is responsible for its unique odor and neuropsychological effects. Crocin mainly contributes to the coloring of saffron and is widely used as a food dye [11]. Saffron production consists of several phases: harvest, collection, transport, drying, packaging, and storage [35].

Flower harvesting and stigma separation are time-consuming; for instance, collecting 1000 flowers takes approximately 45–55 min; another 100–130 min are required to remove the stigmas for drying. [8]. The flowering period lasts from 15 to 25 days, and harvesting begins before dawn, as the flower life is extremely short (from 20–24 h, bud to anthesis) [36]. The collection is completed carefully by cutting the bottom of the corolla. At that point, flowers are transferred to baskets or bags to the processing areas, avoiding excess pressure and deformation of the floral stigmas. Cutting of the flowers is done before the tepals open to prevent them from wilting in the sun (losing their color and taste) [2,17,28,37,38,39]. Traditional harvesting contaminates the product with microorganisms from soil, dust, and sewage. Ethylene oxide decontamination is a conventional method for spice sterilization. However, the process is not reliable due to possible toxic waste [40].

## 3. Post-Harvest

After harvest, there follows a process called *mondatura*, which is carried out in sheds or houses. This phase consists of opening the flower and stigmas are separated from tepals and stamens, for which the stigma is cut at the base of the filaments and the style is removed. Mondatura is done by hand with clean gloves, although machinery with ventilation is used to improve flower separation [2,17,28,38,39]. Stigma harvesting and separation is done manually and is time consuming. Winterhalter and Straubinger [8] have reported that the collection of 1000 flowers take approximately 45–55 min, and another 100–130 min are required to remove the stigmas for drying. This represents around 370–470 h of work to produce 1 kg of dry stigmas. In addition to the time used, the large amounts of tepals (92.6%) obtained become biological waste or useless material [41].

The drying process converts *C. sativus* stigma into saffron and imparts a complex taste due to the combination of flavor, aroma, and color [3,8]. Although fresh stigmas produce a minimal odor, this characteristic appears during drying and storage [42]. The quality of is described by the solubility of its metabolite and the concentrations of picrocrocin, crocins, and safranal [43].

### 3.1. Drying

The drying process is the most important post-harvest step for converting *C. sativus* stigmas into saffron; a valuable aromatic spice with a retail market price of five euros per gram [44,45]. During this process, physical, chemical, and biochemical achieve the desired final properties of saffron, including aroma formation, chemical stability, and antimicrobial activity [46]. Temperature regulation and humidity maximize quality attributes until the stigmas lose 80% of their weight; this treatment includes such crucial points as not washing the stigmas (crocins are soluble in water and could be leached) and abstaining from light exposure (as it affects apocarotenoids). It is recommended to maintain a temperature of 35–45 °C with a relative humidity not exceeding 50%. The drying time will depend on the place, temperature, relative humidity, and raw material load. Drying variables differ between countries and based on experience, available resources, and the climate for each region; this results in saffron quality variations [17,38,47]. Dehydration conditions affect moisture content and the presence and concentration of secondary metabolites; however, the deterioration rate depends on moisture content, temperature, light exposure, oxygen, and enzymatic activity [48]. A high drying temperature decreases the number of bacterial species. The most widely recognized organisms in saffron contamination are those belonging to Bacillaceae family [35].

#### 3.1.1. Conventional Methods

Producers use various drying processes, including sun-drying or drying in the shade under air-ventilation conditions at room temperature (India, Iran, and Morocco), moderate temperature (Greece and Italy), or high temperature (Spain). In India, stigmas are dried in the sun for 3–5 days until their moisture content is reduced to 8–10%. In Morocco, they are placed on a cloth in a thin layer and dried in the sun for several hours or in the shade for 7–10 days. In Italy, they are placed on a 20 cm sieve on charcoal (oak wood) and turned over in the middle of the process (uniform drying). The process is considered a failure when the stigmas do not crumble (i.e., when they exhibit elasticity when pressed between the fingers), with a humidity between 5–20%. In Greece, the stigmas and part of the stamens are spread on trays with shallow layers (4–5 mm) of 40–50 cm with a silk fabric bottom. During the first hours, the process is maintained at 20 °C and then increased to 35–45 °C. Drying ends when a moisture content of 10–12% is reached [49]. Traditional Spanish methods operate at temperatures ranging from 75–121 °C for 28–55 min. In Australia, stigmas are dried using food dehydrators that use hot air flow at temperatures ranging from 40–55 °C (saffron ISO category) [50]. The most conventional industrial process is to use hot air; however, this has low efficiency and high energy costs, which negatively impact quality attributes due to an extended processing time [46]. This technique has shown that higher drying temperatures between 90–110 °C increase the dye strength compared to treatments at lower temperatures (70 °C). This is related to the secondary metabolites detected in chromoplasts, meaning that elevated temperatures could facilitate the release of crocins [51]. Nevertheless, Carmona et al. [49] showed that saffron stigmas dried following the traditional process (sun-dried) exhibit greater coloring strength and better texture than in hot air processes; this result is consistent with Yao et al. [52], who noted that naturally dried saffron shows a compact microstructure (low surface dehydration for a long time); in contrast to high-temperature dehydration, which results in a porous structure, micro surface stress, and irregular bumps that lead to high moisture transfer and a high diffusion rate to the medium. In general, this shows that traditional drying is preferable for a better taste [46]. Dry saffron contains minerals (5–7%), lipids (5–8%), proteins (12–13%), reducing sugars (20%), pentosans (6–7%), gums and dextrins (9–10%), and essential oil (0.3%) [53]. Low temperature drying has a small investment cost, but prolonged time and uncontrolled drying conditions result in compound deterioration (related to color) and quality loss [46,54].

#### 3.1.2. Non-Conventional Methods

Traditional drying involves temperature and mass transfer phenomena within the material [55]. However, several studies have investigated non-conventional drying techniques, including infrared [55,56,57] and microwave radiation [57,58,59], electric ovens [52,58,59] lyophilization [53,57], hot air [50] and vacuum ovens [52,57,59]. Chen et al. [57] determined that, compared to other methods, such as those involving vacuum or electric ovens or microwave o infrared radiation, freeze drying maintained a color (bright and original) and cell structure (i.e., the skeletal structure of the stigma apical cells) very similar to those of the fresh product. It also maintains a high concentration of total crocins. However, this technology incurs a high cost and energy consumption. Yao et al. [52] described the vacuum-dried (100 °C) saffron microstructure, which showed very marked elongated protuberances. Damage to saffron’s surface structure could be associated with moisture transfer during the process; when the temperature reached or approached the boiling point, the moisture in the material was in a state of maximum evaporation, damaging its histological structure.

The drying process ends when saffron stigmas are unbroken and still have a certain elasticity; the final product must maintain a maximum value of 12%, according to ISO Standard 3632 [60]. Regardless of the drying method, it involves various modifications to the color, taste, and aroma of the aromatic spice. Meanwhile, drying conditions affect the chemical profile of the most expensive aromatic species in the world [61]. The higher the drying temperature, the lower the number of bacterial species [35].

### 3.2. Storage

Saffron should be stored in tightly closed containers protected from light immediately after drying. The finished product is available as filaments or powder (deep orange to reddish-brown); this presentation can be easily adulterated. Standard saffron (genuine or unadulterated) has a pleasantly spicy, bitter taste, and a strong odor [38]. Storage favors oxidative or hydrolytic decomposition of secondary metabolites (crocins and picrocrocin), and the rate at which the reactions take place will depend on relative humidity, temperature, and light exposure [47,62,63]. Drying and storage processes represent a critical step toward obtaining saffron with suitable quality, but inappropriate storage can significantly affect its properties [47].

## 4. Theories for the Generation of the Main Compounds in *C. sativus* and Degradation in Saffron

In the food industry, saffron is employed for its sensory attributes, such as its coloring, which is related mainly to crocins, such as mono-glycosyl esters or di-glycosyl polyene. Flavor is related to picrocrocin (terpenoid glycoside), and odor mainly by safranal (a volatile compound). Crocin has been identified as the primary constituent of saffron, making up a proportion of 6–16%. Picrocrocin accounts for between 1 to 13%, and safranal accounts for approximately 2% [64]. The concentration of these secondary metabolites are influenced by the variety, *C. sativus* culturing conditions, and saffron processing methods, specifically whether dried or roasted [5,51]. The saffron drying process affects the chemical profile of aromatic species [61]. Carotenoids are the main precursors of quality compounds. Zeaxanthin can be excised by two routes: enzymatic (using lipooxygenase, xanthine oxidase, phenoloxidase and peroxidase) and non-enzymatic (using photooxygenation, autooxidation and thermal degradation) [35,65]. Various studies have proposed possible theories for the generation in *C. sativus* and degradation of saffron’s main volatile compounds (i.e., their changes in concentration during drying).

### 4.1. Enzymatic Theory in C. sativus

Apocarenoid biosynthesis occurs in *C sativus* stigmas and involves two pathways: the mevalonic acid pathway, which occurs in the cytoplasm, and a non-mevalonic acid pathway, which takes place in plastids and provides carotenoid precursors [66]. Moreover, *C. sativus* metabolites responsible for the color, flavor, and aroma are produced from carotenoids by enzymes that are synthesized in the plastids and then transported to vacuoles for storage [67]. Figure 2 shows how *C. sativus* stigmas quality compounds are obtained from β-carotene, which becomes zeaxanthin by β-carotene hydroxylase (BCH) [68]. Then zeaxanthin is hydrolyzed by the carotenoid cleavage dioxygenase (CCD). Symmetrical breaking of the double bond between the carbons C7-C8 and C7’-C8’ of both ends is performed to generate dialdehyde crocetin and two molecules of 3-OH-cyclocitral (HTCC) [65]. Dialdehyde crocetin is the substrate of aldehyde dehydrogenase (ALDH) used to produce crocetin, which becomes the principal color component (crocin) using the enzyme uridine diphosphate glycosyltransferase (UGT). For bitter taste, the compound generated after zeaxanthin cleavage by CCD, HTCC, is obtained [10,11]. The next step is the glycosylation of HTCC through the action of UGT (produced in the chromoplast of the stigma) to produce picrocrocin. Sereshti et al. [69] described two ways of producing safranal: (1) through an acid medium, basic or high temperature safranal is obtained directly; (2) through the action of the UGT, HTCC produces a glycosylated compound called picrocrocin; subsequently, picrocrocin loses its glycosidic residue through the -glucosidase (-GS) and again becomes aglycone (HTCC); or safranal by heat treatment or hydrolysis of acid or alkaline, losing a water molecule [35,59,69,70]. Martí et al. [68] 2020, described the production of safranal from spontaneous deglycosylation of picrocrocin in *C. sativus*.

### 4.2. Excision of Carotenoids by CCD

Another hypothesis suggests there might be a second way to justify the presence of saffron volatile compounds from C8, C9, C10, C11 and C13 [71]. Is based on carotenoid cleavage by dioxygenases (CCD1, CCD4) in multiple double bonds to generate a series of apocarotenoids in plants [11]. Figure 3 shows that the breakage between C6 and C7 produces a C9 aromatic compound (isophorone). The C7-C8 bond breakdown produces C10 components (-cyclocitral). C11 and C13 compounds were obtained by hydrolyzing between C8-C9 and C9-C10, generating 1-homocyclocitral and an ionone, respectively. The generation of these volatile compounds is produced by the conversion of carotenoids into aromatic compounds through three phases: oxidative cleavage, enzymatic transformation, and acid catalyzed conversions [65].

### 4.3. Crocetin Ester Cleavage to Form Volatile Compounds

Another theory proposes that crocetin esters obtained by enzymatic action may be precursors of safranal though enzymatic action or drying at high temperature. This mechanism was proposed by Carmona et al. [72], who explained that high-temperature dried saffron has no active enzymes but that, at room temperature, it could maintain enzymatic activity over the drying time, i.e., picrocrocin would be the precursor of safranal (active enzymes). Once the saffron is dried, safranal, isophorone and other volatiles are generated from the thermal degradation of trans crocetin esters. This investigation determined that high amounts of eucarvone were found in ancient saffron samples. According to this degradation process, eucarvone loses a molecule of CO and forms C9 (-isophorone and 2-hydroxy-isophorone). Volatile formation from high-temperature crocetin degradation (210 °C) could occur through 4,5,6,7-tetrahydro-7,7-dimethyl-5-oxo-3H-isobenzofuranone (TDOI) and HTCC, as shown in Figure 4.

## 5. Determination of Volatile Chemical Compounds in Saffron

Saffron contains more than one hundred of volatile chemical compounds that appear once stigmas are dehydrated and aroma precursors are transformed [71,73]. Saffron aromatic compounds are classified into (1) mono and sesquiterpenes derived from the isoprenoid synthesis pathway; (2) phenylpropanoids and benzenoids formed in shikimic acid pathway; and (3) compounds from the enzymatic conversion of lipids by β-oxidation. Moreover, some volatile compounds are produced by shortening or modifying the molecules’ skeleton [74]. Safranal is the major component (30–70%) responsible for the saffron aroma, representing 0.001–0.006% of the dry matter [75]. It is synthesized from picrocrocin by an enzymatic process and/or dehydration (either by producing an HTCC intermediate or directly by thermal degradation). However, there is evidence thar this conversion can be performed at low pH values (4). The minor volatile compounds detected in saffron are classified as safranal analogs (compounds related to isophorones with C9 and C10), norisoprenoids (compounds related to isophorones with C13), and saturated linear hydrocarbons [5,76,77]. Saffron minority aromatic molecules are formed from safranal degradation and lipophilic carotenoids [78]; other important compounds are glycosylated after undergoing hydrolysis with a volatile profile [79]. Table 1 shows major detected components of saffron from diverse geographical origins, as obtained by various extraction and detection techniques. Safranal represents the principal constituent of almost all reports. A correlation between safranal concentration and minority compounds with other factors, such as harvest season, dehydration temperature, and storage time/conditions, can be observed [77].

Geographical origin is considered an essential factor in the concentration of different volatile molecules detected, since the producing countries use different post-harvest processes, specifically for dehydration [57,71]. However, safranal is the majority compound, regardless of its origin (Spain, Greece, Iran, Morocco, Italy) [19,45,80]. Karabagias et al. [19] determined the total concentrations of volatile compounds in saffron from various origins, finding that they were arranged in the following order: Morocco > Iran > Greece > Spain. Ghanbari et al. [81] evaluated the variation of volatile compounds in saffron corms from diverse regions; their data suggest that origin can affect saffron quality. Fancello et al. [35] described the relationship between safranal concentration and the drying method and reported three types of drying: high temperature in a drying oven (Spain), in sunlight at room temperature (India, Iran, Morocco), and in the shade (Greece, Italy). The results showed that the concentration of safranal detected increased with the increase in drying temperature (oven > sun > shade). Chen et al. [57] showed that the structure, quality, and aromatic compounds of saffron change when subjected to different drying methods, including vacuum, oven, microwave and infrared radiation, and lyophilization. The results showed that microwaves could be the most suitable technique for obtaining volatile compounds with a minimal drying period. Furthermore, the optimal storage time was reported by Nenadis et al. [82] using proton-transfer-reaction mass spectrometry (PTR-MS) to determine volatile compounds in fresh and stored saffron. The analysis showed that volatile compounds fingerprints change over time, and safranal specifically suffered a progressive decrease. Another critical factor determined by this research was that particle size affects the concentration of volatile compounds. Grinding increases this concentration through the release of organic compounds produced by the breakdown of tissues when pulverized.

Besides safranal, other important volatile compounds in saffron were determined include isophorone, ketoisophorone, HTCC, and isophorone isomers. Anastasaki et al. [45] used GC-MS/FID to determine the major compounds in Italian and Spanish samples: safranal, 4-hydroxy-2,6,6-trimethyl-3-oxocyclohexa-1,4-diene-1-carboxaldehyde, HTCC, isophorone, and dihydrooxophorone. The same compounds were determined in samples from Greece and Iran but in a different order of concentration. Aliakbarzadeh et al. [86] used GC-MS, 77 volatile compounds from Iranian saffron, finding 11 biomarkers, including 9 secondary metabolites (β-Isophorone, Phenylethyl alcohol, α-Isophorone, Ketoisophorone, Dihydrooxophorone, Safranal, 2,6,6-Trimethyl-4-oxo-2-cyclohexen-1-carbaldehyde, 2,4,4-Trimethyl-3-carboxaldehyde-5-hydroxy-2,5-cyclohexadien-1-one and HTCC), a primary metabolite (linoleic acid), and a long-chain fatty alcohol (nonacosanol). Azarabadi & Özdemir [85] used SPME to extract and determined the significant compounds from Iran, including acetic acid, 2-(5*H*)-furanone, isophorone, 4-ketoisophorone, 2,6,6-trimethyl-1,4-cyclohexanedione, eucarvone, and safranal. In the same way, Carmona et al. [80] determined high concentrations of acetic acid, isophorone, and dihydrooxophorone in samples from Iran and Morocco as markers to distinguish their origin. Moreover, zero or minimal concentrations of acetic acid were found in Spanish and Greek saffron; this organic acid is produced due to a rapid elimination of water during dehydration. D’Archivio [87] used HS-SPME/GC-MS to determine the seven main chemical aromatic components of saffron (safranal isomer, isophorone, 4-ketoisophorone, dihydrooxophorone, 7,7-dimethylbicyclo (4.1.0) hept-3-ene-2,5-dione, 4-hydroxy-2,6,6-trimethyl-3-oxocyclohexa-1,4-diene-1-carbaldehyde and HTCC). Chemical composition of volatile compounds depends not only on the plant, post-harvest factors, or origin, but also on the extraction method [73]. Consequently, the preparation, extraction, and characterization procedures can vary the concentration of the compounds of interest [86]. Conventional techniques are the most widely used, but the significant disadvantages (reduced efficiency, high energy consumption, long extraction time, and degradation of some compounds) open the door for novel extraction methods [88]. Today, gas chromatography is a suitable analytical technique for detecting volatile compounds. These singular analytical methods could be considered to assess the quality of saffron and thus included in the ISO technical specifications [70,73,77]

## 6. Saffron’s Aroma Descriptors

The entry of an aroma into the nose during inhalation stimulates olfactory receptors through the orthonasal route; simultaneously, aromatic compounds are released in the mouth by decomposition and incorporation into saliva. The aroma is distributed between saliva and oral cavity, while scented air is transferred to the pharynx, followed by an interaction with epithelium olfactory receptors that transduces sensory signals to the brain [89]. Food compounds contribute to the overall aroma; therefore, it is crucial to separate the less or odorless ones from the intensely active odor compounds [79]. There is no direct correlation between aromatic compounds concentration and the perception intensity and/or odor threshold [89]. Currently, saffron is very popular as a colorant due to its aromatic and flavoring properties; however, freshly harvested saffron is odorless, as odoriferous substances only develop a pleasant odor during the drying process [35,79,90]. Safranal and isophorone deserve particular attention as key odorants in saffon [90,91]. Amanpour et al. [92] determined that the volatile compounds that contribute to the saffron aroma can be formed through (i) lipid oxidation; (ii) Strecker degradation/Maillard reaction; and (iii) glycosides hydrolysis from a non-carotenoid precursor. The bitter taste and iodine form fragrance (like that of hay) are related to picrocrocin and safranal, respectively; picrocrocin is the compound responsible for saffron’s bitter taste, together with kaempferol and its derivatives. Nevertheless, norisoprenoids represent components that impart saffron’s aromatic properties [5,70].

The volatile compound chemical profile is obtained by GC-MS analysis; however, to locate and classify aromatically important odorants, olfactometric studies or PTR-MS are required [93]. Table 2 shows the main volatile compounds of and the respective aromatic notes for each one. Maggi et al. [94] determined seven aromatic notes in Spanish saffron extracts using (GC–SM) and gas chromatography-olfactometry (GC–O), obtaining vegetable (grass), flower (pungent flower, fresh flower), caramel, freshly cut grass, spicy (typical saffron aroma), and citrus notes, which are concordant to those described by Amanpour et al. [79] in Iranian saffron. These authors propose nine aromatic descriptors of saffron: saffron, flowery, spicy, fresh-cut grass/grassy, vegetable, citrus, caramel-like, mushroom/earthy, and vinegary. The first four descriptors possess a high intensity while the last five show a low intensity. Only two notes differed from those of Maggi et al. (mushroom/earthy and vinegary).

Saffron’s characteristic aroma comprises a complex combination of several scents and is not just an isolated active aromatic compound [95]; safranal has been determined to be the most important essential aromatic compound in saffron, but the aroma emitted is due to at least twenty different volatile compounds [91]. Guclu et al. [95] reported that saffron’s key odorants include safranal, 4-ketoisophorone, 2-hydroxy-4,4,6-trimethylcyclohexa-2,5-dien- 1-one and dihydrooxoforone. The combination of these compounds is responsible for the typical aroma of the spice. The volatile compounds mentioned above, as well as four other components (isophorone, isophorone isomer, safranal isomer, and HTCC), are saffron’s important aroma components [79]. Recently, safranal and isophorone were established as aroma and flavor descriptors contributing to the herbaceous notes, a saffron hallmark [40,83].

It is important to emphasize that isophorone has two anomers that produced distinctive notes. α-isophorone (a transparent liquid compound) denotes a minty smell [96,97], while β-isophorone has floral and grass descriptors [96]. The α and β isomers of isophorone, safranal, ketoisophorone, 2,2,6-trimethyl-1,4-cyclohexanedione, 2,6,6-trimethyl-4-oxo-2-cyclohexen-1-carbaldehyde, HTCC, linoleic acid, nanocosanol and β-phenylethanol, were determined to be the aromatic biomarkers of Iranian saffron [86]. β-Phenylethanol possesses a pleasant flower or green roses aroma with antimicrobial properties [79,86]. The aromatic composition of saffron from Teruel, Spain was evaluated by GC-O, and at least twenty different aroma molecules were identified, with safranal, 2,3-butanedione, hexanal, *E*-2-nonenal, and an unidentified burnt curry smell highlighted. Saffron’s toasted aroma profile is related to an increase in pyrans and furans by the Maillard reaction, which generates their intensification.

The absence of these compounds in fresh species that generates by the drying process reinforces this theory. Another compound related to roasting is megastigmane; its formation supports the hypothesis that crocetin, zeaxanthin, and crocetin esters degrade, and that their hydrolytic products contribute to flavor and aroma [5]. Another saffron important odorant is linalool, which contributes fresh and floral notes [74,79]. Aromatic compounds can also be used as freshness markers, for example, by looking at the ratio of ketoisophorone to safranal as an aging indicator [5]. Both components contributed spicy and floral notes within a year after harvesting. However, during prolonged storage (>3 years), these compounds degrade, increasing caramel, citrus, and vegetable notes [82,94].

## 7. Conclusions

The high commercial value of saffron is related to its complicated production process (harvest, drying, storage), that still requires manual operations, as well as to the low yield of the final product. It is important to consider that the post-harvest process, in particular the drying treatment, is fundamental to the generation of the volatile compounds related to saffron’s quality. Treatments can vary from country to country and from producer to producer and can also affected saffron’s chemical composition. Chemical characterization of saffron’s derivatives include compounds such as safranal, picrocrocin, linalool, crocin, ketoisophorone, isophorone, crocetin, among others. The identification of key components in saffron can help on the speciation and will provide indicators to assess the purity and quality of the final product.

## Figures and Tables

**Figure 1 molecules-26-06954-f001:**
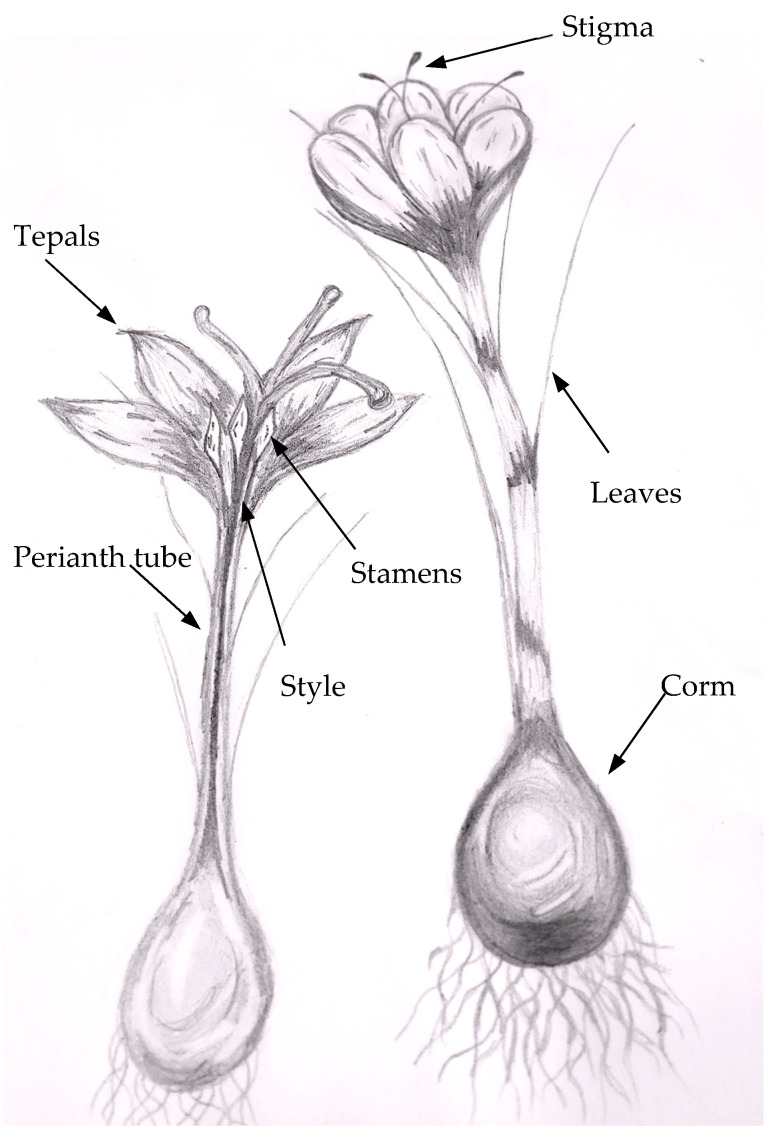
*Crocus sativus* L. plant parts.

**Figure 2 molecules-26-06954-f002:**
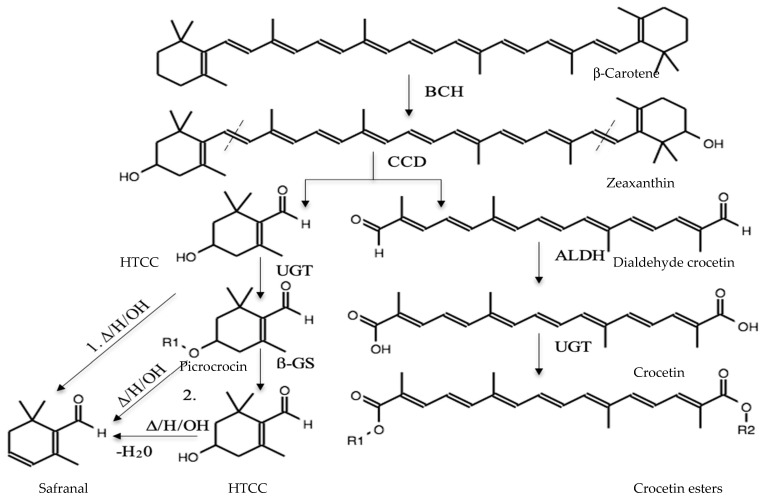
Enzymatic pathway in *C. sativus* and saffron chemical degradation of aromatic compounds from zeaxanthin.

**Figure 3 molecules-26-06954-f003:**
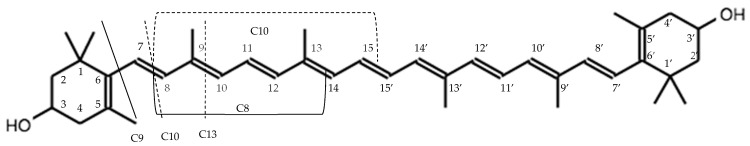
Zeaxanthin degradation by CCD, compounds generation in C9 (4-ketoisophorone, isophorone, dihydrooxophorone, 2-hydroxy-isophorone); C10 safranal similar structures (HTTC, 5,5-dimethyl-2-methylene-3-cyclohexene-1-carboxaldehyde, 3,3,4,5-tetramethylcyclohexan-1-one, 2,6,6-trimethyl-4-oxo-2-cyclohexen-1-carbaldehyde, -cyclocitral, 2,4,4-trimethyl-3-carboxaldehyde-5-hydroxy-2,5-cyclohexadien-1-one); C11 (-homocyclocitral) and C13 (ionone, 2-cyclohexen-1-ol, 3-(1-buten-1-yl), 2,4,4-trimethyl, megastigma-4,6(*E*),8(*E*)-triene).

**Figure 4 molecules-26-06954-f004:**
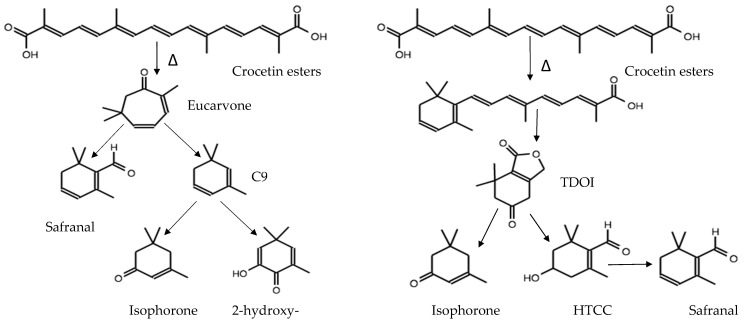
Saffron volatile compounds generation from crocetins esters by thermal degradation.

**Table 1 molecules-26-06954-t001:** Main volatile compounds of saffron quantified by GC obtained from different extracts and geographical origins.

Origin and Type of Extract	Compound Name	Concentration		Technique	Reference
Iran (NaCl aqueous extract)	2,2,6-Trimethyl-1,3-cyclohexadiene-1-carboxaldeyde (Safranal)	7650	μg/kg	HS-SPME/GC-MS	[19]
	1,5,5-Trimethyl-6-(2-butenylidene)-cyclohexene, (Megastigma-4,6(*E*),8(*E*)-triene)	1113			
	5,5-Dimethyl-2-methylene-3-cyclohexene-1-carboxaldehyde	689.5			
	3,7-Dimethyl-octa-1,6-dien-3-ol (Linalool)	587.5			
	2,4-Dimethylbenzenecarboxaldehyde	130.5			
Spain (Aqueous extract)	Safranal	4510	μg/kg	HS-SPME/GC-MS	[19]
	3,5,5-Trimethyl−3-cyclohexen-1-one (β-isophorone)	924.5			
	5,5-Dimethyl-2-methylene-3-cyclohexene-1-carboxaldehyde	357.5			
	Linalool	120.8			
	2,6,6-Trimethyl-2-cyclohexene-1,4-dione (4-ketoisophorone)	113.8			
	5-Hydroxy-2,5-cyclohexadien-1-one-2,4,4-trimethyl-3 carboxaldehyde	110.2			
Greece (Aqueous extract)	Safranal	6450	μg/kg	HS-SPME–GC-MS	[19]
	2,4,5-Trimethyl-benzaldehyde	1380			
	β-Isophorone	747			
	Megastigma-4,6(*E*),8(*E*)-triene	715.5			
	Linalool	286			
	5,5-Dimethyl-2-methylene-3-cyclohexene-1-carboxaldehyde	285.5			
	2,6,6-Trimethyl-2,4-cycloheptadien-1-one (Eucarvone)	243			
Morocco (Aqueous extract)	Safranal	31,710	μg/kg	HS-SPME–GC-MS	[19]
	5,5-dimethyl-2-methylene-3-cyclohexene-1-carboxaldehyde	2890			
	1-Methyl-3-(1-methylethyl)-benzene (m-Cymene)	1960			
	β-isophorone	779			
	Dodecene	492			
	3,5,5-Trimethyl-2-cyclohexen-1-ol (Isophorol)	452			
Khorasan, Iran (Methanol-ethy-acetate ultrasonic extraction)	Safranal	26.29	%	GC-MS	[76]
	Bicyclo [3,2,0] hept-2-ene-4-ethoxy-endo	5.69			
	4-Hydroxy-2,6,6-trimethyl-1cyclohexene-1-carboxalde-hyde (β-Homocyclocitral)	4.44			
	Linoleic acid	4.77			
Khorasan, Iran. (aqueous-dichloro methane)	Safranal	2168	μg/g	SAFE-GC-MS	[79]
	2-(1,1-Dimethylethyl) phenol	1432			
	2-Ethoxy-5-methoxybenzaldehyde	1147			
	3,5,5-Trimethyl-2-cyclohexen-1-one (α-Isophorone)	845			
	4-Ketoisophorone	625			
	3,5,5-Trimethyl-1,4cyclohexanedione (Dihydrooxo phorone)	591			
Sicily, Italy (NaCl aqueous extract)	Safranal	84.38	%	HS-SPME/GC-MS	[26]
	2,6,6-Trimethyl-1,4-cyclohexadiene-1-carboxaldehyde (Safranal isomer)	5.72			
	α-Isophorone	3.96			
	Furfural	1.91			
	2,4-Dimethylbenzenecarboxaldehyde	1.2			
Eskisehir, Turkey (Aqueous microdistiller)	Safranal	64.1	%	MD/GC-MS	[16]
	α-Isophorone	10.38			
	β-Isophorone	6.4			
	Hexadecanoic acid	3.3			
	4-(2,2,6-Trimethyl-cyclohexan-1-yl)-3-buten-2-one (β-Ionene)	1.25			
	Safranal isomer	1.13			
Safranbolou, Turkey (Aqueous microdistiller)	Safranal	49.33	%	MD/GC-MS	[16]
	α-Isophorone	16.25			
	β-Isophorone	8.38			
	Hexadecanoic acid	4.05			
	β-Ionene	2.8			
	4-Ketoisophorone	2.5			
	3,3,4,5-Tetramethylcyclohexan-1one	2.28			
Eskişehir, Turkey (Aqueous microdistiller)	Safranal	77.9	%	MD/GC-MS	[16]
	α-Isophorone	13.5			
	β-Isophorone	2.2			
	Safranal isomer	1.9			
	4-Ketoisophorone	1.2			
Qaen, Iran (Methanol-acetonitrile ultrasound-assisted extraction)	Hexadecanoic acid	25	%	UAE/DLLME/GC-MS	[83]
	Safranal	16.8			
	Tetradecanoic acid	14.3			
	5,5-Dimethyl-2-methylene-3-cyclohexene-1-carboxaldehyde	7.5			
	α-Isophorone	4.9			
	4,4-Dimethyl-2-cyclopenten-1-one	4.9			
Kerman, Iran (Ground agitated with water headspace)	2(5*H*)-Furanone	691.8	ppb	TR/TOF/MS	[84]
	Safranal	610.1			
	Acetic acid	566.3			
	Isobutanal	451.8			
	Biogenic aldehyde	272.6			
	4-Ketoisophorone	161.2			
	Acetaldehyde	130.4			
	α-Isophorone	106.6			
Hangzhou, China (Microwave drying)	Safranal	20.54	%	GC-MS	[57]
	Dihydrooxophorone	18.98			
	4-Ketoisophorone	16.11			
	α-Isophorone	14.57			
	2,3-Dimethoxytoluene	11.68			
Hangzhou, China (Oven drying)	Safranal	15.73	%	GC-MS	[57]
	2,3-Dimethoxytoluene	13.94			
	Dihydrooxophorone	13.18			
	4-Ketoisophorone	11.73			
Hangzhou, China (Infrared drying)	Safranal	22.18	%	GC-MS	[57]
	Dihydrooxophorone	16.88			
	4-Ketoisophorone	16.09			
	α-Isophorone	15.68			
	2,3-Dimethoxytoluene	12.83			
Hangzhou, China (Freeze drying)	4-Ketoisophorone	28.2	%	GC-MS	[57]
	Safranal	10.31			
	2,3-Dimethoxytoluene	7.19			
	3-Methyl-2-cyclohexen-1-one	5.37			
	α-Isophorone	5.17			
Iran (Microextraction by fiber polyacrylate)	Safranal	59.32	%	SPME/GC-MS	[85]
	Isophorone	11.48			
	4-Ketoisophorone	10.66			
	Dihydrooxophorone	8.35			
Iran (Microextraction by fiber polydimethyl-siloxane)	Safranal	49.64	%	SPME/GC-MS	[85]
	Acetic acid	9.49			
	4-Ketoisophorone	8.72			
	Isophorone	8.2			
Iran (Microextraction by fiber carboxenpoly-dimethylsiloxan)	Safranal	55.51	%	SPME/GC-MS	[85]
	Isophorone	14.95			
	4-Ketoisophorone	10.52			
	Isophorone isomer	10.05			

**Table 2 molecules-26-06954-t002:** Saffron volatile chemical compounds with their respective odor descriptor.

Compound	Odor Descriptor	Reference
(*E,E*)-2,4-Decadienal	Fatty, deep-fried, fried fat	[91,95]
(*E,E*)-2,4-Nonadienal	Rancid oil	
(*E,Z*)-2,6-Nonadienal	Cucumber, sweet	
(*E*)-2-Nonenal	Melon, aldehydic	
(*E*)-Geranyl-acetone	Magnolia, green	[26]
(*Z*)-2-Nonenal	Green, metallic	[91,95]
1-Octen-3-one	Mushroom, earthy	
1-Tetradeacanol	Coconut	[26]
2-Acetyl-1-pyrroline	Nutty, popcorn	[95]
2-Ethyl-hexanol	Rose Green	[26]
2-Hydroxy-3,5,5-trimethylcyclohex-2-ene-1,4-dione	Flower, woody	[26,94]
2-Hydroxy-4,4,6-trimethyl-2,5-cyclohexadien-1-one	Caramel, Saffron, stale, dried hay	[94,95]
2-Hydroxyisophorone	Woody	[26]
2,2-Dimethyl-cyclohexane-1-carboxaldehyde	Flower	[94]
2,3-Butanedione	Butter, cream, cream cheese	[91,95]
2,4-Dimethyl-benzaldehyde	Almond, spice	[26]
2,4,5-Trimethyl-benzaldehyde	Floral, violets	
2,5-Dimethyl-benzaldehyde	Spice	
Isomer of safranal	Caramel	[94]
3-[(*E*)-But-1-enyl]-2,4,4-trimethyl-cyclohex-2-enol	Citrus	
3-Hexen-2-one	Grass, geranium	[91,95]
3-Methylbutanoic acid	Cheese, rotten, sour, dried fruit	[95]
3-(Methylthio) propanal	Baked potato	
4-Hydroxy-2,6,6-trimethyl-3-oxocyclohex-1-en-1carboxaldehyde	Citrus, vegetable	[94]
HTCC	Green, Spicy, saffron, green	[26,79,94,95]
4-Hydroxy-2,6,6-trimethyl-3-oxocyclohexa-1,4-diene-1carboxaldehyde	Fresh cut grass	[94]
4-Hydroxy-3,5,5-trimethyl-2-cyclohex-1-one	Flower, vegetable	
4-ketoisophorone	Saffron, vegetable, musty, woody	[26,79,94,95]
6-Methyl-5-hepten-2-one	Clove, spicy, green, citrus like	[26,91,95]
Acetic acid	Vinegar, acidic	[91,95]
Butyric acid	Cheese	
Carvone	Mint	[26]
Dihydro-β-ionone	Earty, woody	
Dihydrooxophorone	Vegetable, saffron	[26,79,94,95]
Eucalyptol	Eucalyptus	[26]
Furaneol	Cotton Candy, strawberries	[91,95]
Furfural	Bread, almond	[26]
Heptanal	Fat, rancid	
Hexanal	Grass	[91,95]
Homofuraneol	Cotton candy	
Isophorone	Saffron, herbal, flower peppemint, woody, hay	[26,79,91,94,95]
Isophorone-4-methylene	Citrus	[94]
Isovalerianic acid	Cheese	[91]
Limonene	Green, citrus	[26]
Linalool	Floral, honeysuckle	[79,95]
Octanal	Lemon, fat, green	[26,91,95]
Oxo-β-cyclocitral	Caramel	[94]
Phenol	Phenol	[26]
Phorone	Geranium	
Rose oxide	Rose, flower	
Safranal	Saffron, tea, herbal	[26,79,91,94]
Teroinolene	Fresh, pine	[26]
α-Cyclocitral	Green	
α-Pinene	Pine, terpenin	
α-Terpinene	Lemon	
β-Cyclocitral	Mint	
β-Ionone	Floral	
-Phenylethanol	Roses, floral, flower	[79,91,94,95]
-Pinene	Pine, resin	[26]
-Terpinene	Turpenin	

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
