# Peer review of "The Relation between Drying Conditions and the Development of Volatile Compounds in Saffron (Crocus sativus)"

_molecules, 2021, doi:10.3390/molecules26226954_

Round 1

Reviewer 1 Report

Dear authors,

This review is informative and is meaningful, but it is a little messy in structure and the main line is not clear enough. Please reorganize the structure of the manuscript sections.

The section number is confusing, such as 2.3 Post-harvest, followed by 3.1.Drying.

What is the difference between ‘Section 4. Saffron aroma compounds and its pathways’ and ‘Section 5 Saffron volatile organic compounds (VOC)’?

In my opinion, the VOC may be natural scent volatiles released from living plant parts, for instance, as determined by HS-SPME/GC-MS. However, in Table 1, different extracts (such as UAE, or MD) are also included.

In addition, the VOCs, if it means natural composition, should be introduced following sections ‘the Saffron as an aromatic spice, Botanical characterisctics’. Also is the ‘section 6. Aroma descriptors.’

What is the aim of Table 1, for evaluating the influences of geographical origin, or for comparing different extraction methods?

Section 4. Saffron aroma compounds and its pathways

What does the pathway mean?

Does it mean the biosynthesis pathways in living plant (for instance, Shikimic acid pathway) or the chemical degradation pathway during the processing? Please clarify it.

There are Grammatical errors in sentences throughout the manuscript. Please check it carefully. For example, Line 436 -438. ‘Is important to consider that post-harvest and specially drying treatments are important to the generation of volatile compounds that can improve saffron overall quality’

The conclusion is too simple.

Thank you.

Author Response

This review is informative and is meaningful, but it is a little messy in structure and the main line is not clear enough. Please reorganize the structure of the manuscript sections.

The section number is confusing, such as 2.3 Post-harvest, followed by 3.1.Drying.

ANSWER: The manuscript structure was reorganized, according to your observations. Thank you for improving our paper.

What is the difference between ‘Section 4. Saffron aroma compounds and its pathways’ and ‘Section 5 Saffron volatile organic compounds (VOC)’?

ANSWER: Section 4 and 5 titles were changed

In my opinion, the VOC may be natural scent volatiles released from living plant parts, for instance, as determined by HS-SPME/GC-MS. However, in Table 1, different extracts (such as UAE, or MD) are also included.

ANSWER: The terms C. sativus and saffron were employed to difference the living plant and the dryed plant.

In addition, the VOCs, if it means natural composition, should be introduced following sections ‘the Saffron as an aromatic spice, Botanical characterisctics’. Also is the ‘section 6. Aroma descriptors.’

ANSWER: The terms were specified along the manuscript

What is the aim of Table 1, for evaluating the influences of geographical origin, or for comparing different extraction methods?

ANSWER: The aim was to evaluate the influence of geographical origin and its different post havest processing.

Section 4. Saffron aroma compounds and its pathways

What does the pathway mean?

Does it mean the biosynthesis pathways in living plant (for instance, Shikimic acid pathway) or the chemical degradation pathway during the processing? Please clarify it.

ANSWER: Your observation is very valuable and considered in section 4.

There are Grammatical errors in sentences throughout the manuscript. Please check it carefully. For example, Line 436 -438. ‘Is important to consider that post-harvest and specially drying treatments are important to the generation of volatile compounds that can improve saffron overall quality’

ANSWER: Grammar was chech it

The conclusion is too simple.

ANSWER. Conclusion was improved

Reviewer 2 Report

The manuscript "The relation between drying conditions and the development of volatile compounds in saffron (Crocus sativus)" presents an interesting research topic concerning a popular spice. From a scientific point of view, I recommend its publication after complementation of its introduction section. The authors failed in contextualizing the research issues. Also, my concerns about this paper refer to English grammar. I suggest a deep language revision.

Author Response

The manuscript "The relation between drying conditions and the development of volatile compounds in saffron (Crocus sativus)" presents an interesting research topic concerning a popular spice. From a scientific point of view, I recommend its publication after complementation of its introduction section. The authors failed in contextualizing the research issues. Also, my concerns about this paper refer to English grammar. I suggest a deep language revision.

ANSWER: We thank your valuable observations.

Introduction was complemented and research issues were improved

Grammar and deep language revision was improved.

Round 2

Reviewer 1 Report

Dear authors,

The improvement of the manuscript quality is apparent.

There are still some minor issues to be addressed:

Abstract
Please consider reorganizing the language.

line 124
After the harvest
After harvest

line 125
separating the stigmas are separated from tepals and stamens
separating the stigmas from tepals and stamens

line 129
harvesting and separation it is done manually and time consuming.
harvesting and separation is done manually and is time consuming.

line 452
can also be affected by saffron’s chemical composition
can also affect saffron’s chemical composition? (I'm not sure if you mean this)

As soon as these issues are addressed, the manuscript can be accepted.

Thank you!

Reviewer 2 Report

After reviewing the authors' responses to my comments, I verified that all suggestions were incorporated into the revised version.